# Spatial Landscape Structure Influences Cross-Species Transmission in a Rabies-like Virus Model

**DOI:** 10.3390/microorganisms13020416

**Published:** 2025-02-14

**Authors:** Norma Rocio Forero-Muñoz, Gabriel Dansereau, Francois Viard, Emily Acheson, Patrick Leighton, Timothée Poisot

**Affiliations:** 1Département de Sciences Biologiques, Faculté des Arts et des Sciences, Université de Montréal, Montreal, QC H2V 0B3, Canada; norma.rocio.forero.munoz@umontreal.ca (N.R.F.-M.); gabriel.dansereau@umontreal.ca (G.D.); 2Département de Pathologie et Microbiologie, Faculté de Médecine Vétérinaire, Université de Montréal, Montreal, QC J2S 8H5, Canada; francois.viard@umontreal.ca (F.V.); patrick.a.leighton@umontreal.ca (P.L.); 3Groupe de Recherche en Épidémiologie des Zoonoses et Santé Publique (GREZOSP), Université de Montréal, Montreal, QC J2S 8H5, Canada; emily.acheson@phac-aspc.gc.ca; 4Public Health Risk Sciences Divisions, National Microbiology Laboratory, Public Health Agency of Canada, Saint-Hyacinthe, QC J2S 0H7, Canada

**Keywords:** neutral landscapes, agent-based models, cross-species transmission, rabies

## Abstract

In this study, we simulated biologically realistic agent-based models over neutral landscapes to examine how spatial structure affects the spread of a rabies-like virus in a two-species system. We built landscapes with varying autocorrelation levels and simulated disease dynamics using different transmission rates for intra- and interspecies spread. The results were analysed based on combinations of spatial landscape structures and transmission rates, focusing on the median number of new reservoir and spillover cases. We found that both spatial landscape structures and viral transmission rates are key factors in determining the number of infected simulated agents and the epidemiological week when the highest number of cases occurs. While isolated habitat patches with elevated carrying capacity pose significant risks for viral transmission, they may also slow the spread compared to more connected patches, depending on the modelled scenario. This study highlights the importance of spatial landscape structure and viral transmission rates in cross-species spread. Our findings have implications for disease control strategies and suggest that future research should also focus on how landscape factors interact with pathogen dynamics, especially in those locations where susceptible agents could be more in contact with pathogens with high transmission rates.

## 1. Introduction

The effect of spatial landscape structure on the dynamic of multispecies epidemics has become an important research focus [1,2]. Understanding and estimating how landscape structure shapes pathogen–host interactions can provide critical insights into assessing risk areas—geographic locations where the likelihood of disease transmission or outbreaks is higher. Numerous factors, including habitat availability [3], environmental conditions [4], and social behaviour [5,6], can be associated with the spatial arrangement or pattern of hosts, such as animals or organisms, within an environment across landscapes [7,8].

Landscape ecology studies the configuration of habitat patches, their connectivity, and the presence of physical barriers that shape species’ movements and interactions [9,10,11] and how these factors influence species’ ability to move between areas and the likelihood of encountering one another [12,13]; this, in turn, plays a critical role in the transmission dynamics of pathogens [14,15]. Considering environmental attributes within an “epidemiological landscape” framework can significantly improve epidemiological forecasts’ accuracy, even without detailed local geographic data [16]. Recent studies underscore the importance of integrating landscape information into outbreak management and public health policies [17]. Spatial epidemiology facilitates the analysis of disease patterns and allows for both short- and long-term forecasting [18] by monitoring climate variability, environmental conditions, and their impacts on infectious dynamics [19]. Moreover, it has contributed to understanding the association between habitat loss, reduced species diversity, and the increased risk of infections [20], underlining how landscape structure changes can influence disease transmission.

Pathogen–host interactions are shaped by ecological and environmental factors that influence the likelihood of infectious spillover [21], including landscape structure, host density, and ecosystem connectivity [22,23,24]. Cross-species transmission of viruses plays a significant role in emerging infections, with implications for population structure, species interactions, and ecosystem dynamics. Although it is a biological process [25,26], we can minimise its impact on human and animal health by addressing the factors facilitating cross-species transmission, including vaccination programs, and preventing high-host-density environments and/or multi-species interactions [27]. Consequently, measuring cross-species transmission risk in human–animal interfaces is essential from a One Health perspective [28]. This involves assessing how changes in ecosystems and anthropogenic factors—such as cultural beliefs, socioeconomic conditions, health-seeking behaviours, and human–animal interactions—affect the likelihood of zoonotic spillovers. Such events have meaningful human and animal health implications, as in the spillover of influenza A(H5N1) from birds to mammals [29,30], and may result in biodiversity loss, as seen with amphibian chytridiomycosis [31,32,33,34].

Lyssaviruses, including rabies virus (RABV), are among the pathogens that continue to pose significant risks to both human and animal health despite ongoing vaccination efforts. Rabies is a viral disease that primarily affects mammals, and it is often transmitted through the saliva of infected individuals, typically through bites [35]. The disease is notorious for its high case fatality, and it continues to cause human and animal deaths globally [36]. In North America, raccoons (*Procyon lotor*) are key reservoir species for rabies [37], with spillover often occurring when these animals come into contact with other wildlife species (e.g., skunks, family Mephitidae), as well as with domestic animals and humans [38]. Therefore, rabies is a suitable model for understanding cross-species transmission and exploring the role of landscape structure in these processes. For example, recent studies used agent-based modelling (ABM) to assess the effectiveness of transmission dynamics or control programs in rabies outbreaks [39,40,41].

A crucial next step is to analyse (a) transmission dynamics and (b) landscape effects on cross-species transmission simultaneously. Understanding these two factors in real-world ecological settings is essential, as variations in habitat structure can drive the spread of diseases [42], animal movement patterns [43], and local population densities [44]. For instance, areas with high landscape connectivity could enable the movement of infected individuals and facilitate disease transmission, while areas with lower connectivity may act as barriers, isolating populations and slowing the spread of diseases [45,46]. Neutral landscape models (NLMs) allow for studying the actual influence of landscape structure by simulating various patterns without the complexity of external biotic processes [47]. NLMs have been employed to explore different ecological processes, including species movement [48] and gene flow [49].

These landscape dynamics are particularly relevant for managing diseases like rabies, where spatial structure can influence the effectiveness of traditional control measures. Approaches such as culling and/or vaccination have been used to control rabies outbreaks, but the effectiveness of these strategies might vary based on landscape connectivity. While culling alone has been ineffective, studies suggest that high vaccination coverage (greater than 80%) is necessary for effective control, as shown for high-density fox populations [50,51]. Thus, measuring landscape effects on disease outbreaks can inform more effective control strategies and reduce the potential for widespread transmission.

Accordingly, we simulated biologically realistic agent-based models over neutral landscapes to explore the influence of spatial structure on the transmission of a rabies-like virus in a two-species system.

## 2. Materials and Methods

We generated several neutral landscapes with varying autocorrelation and then built a two-species (i.e., reservoir and spillover) agent-based model to simulate transmission dynamics. Landscape (matrix) composition was simulated using Julia v1.11 through the NeutralLandscapes.jl package. ABMs were run on Python v3.8 using the SamPy package [52]. To provide an analogue for the spread of the raccoon rabies virus variant, we specified the model parameters to approximate population dynamics and demographic parameters of its mammalian reservoirs, raccoons (*Procyon lotor*) and skunks (family Mephitidae).

### 2.1. Landscape Structure: Neutral Landscape Models (NLMs)

We built neutral landscapes (NLMs) [53] with varying levels of autocorrelation inside a grid of 10,246 hexagonal cells, which are based on the Diamond-Square algorithm [54,55]. Briefly, the algorithm assigns random heights to the four corners of a grid. First, the heights of these corners were averaged, and a small random value was added. This average was set as the height of the centre point of the square formed by the corners. Subsequently, the algorithm calculates the average height of the four corners of each diamond shape created between the squares. Then, another random value was added and assigned to the diamond’s centre point.

Autocorrelation measures how comparable carrying capacities [56,57] are distributed within a simulated landscape. High autocorrelations imply that nearby points have similar density values, resulting in smoother transitions and a tendency towards homogeneity. In contrast, low autocorrelations indicate more significant variability and abrupt changes between values (more evident patchiness and heterogeneity) [58,59]. Simulating a gradient of autocorrelation values allowed us to cover various possible processes and relations to neighbouring cells, creating different landscape models with realistic features.

### 2.2. Population Dynamics

We followed the Grimm et al. [60] protocol for creating agent-based models (ABMs). Building on Acheson’s et al. one-species model [39], we considered the existence of two species of agents (i.e., reservoir and spillover), with each agent (representing individuals in our modelling framework) being defined by specific attributes (e.g., age and sex) (further information regarding ABM’s parameters is available in the Appendix A). Additionally, interactions were considered in cases where a susceptible individual encountered an infected one, either within the same species or between species in a defined landscape (NLMs). Simulations were performed using different combinations of intra- and interspecies (spillover and spillback) transmission rates, contrasting them under ten levels of autocorrelation (see section below).

In our ABM, each cell represents a discrete unit that simulates disease transmission dynamics by implementing an SI (susceptible-infected) framework. Agents are categorised as susceptible (S)—those not infected—or infected (I)—those suffering from the viral disease. Spillover species need at least one contact with an infected reservoir to initiate the first case and trigger the outbreak. Interactions among agents are controlled by specific rules that account for proximity and transmission probability, influencing the transition from susceptible to infected states. System dynamics followed population and disease parameters described for raccoons in Acheson et al. [39], including reproduction, dispersal and mortality (both disease-related and non-disease-related).

The first outbreak occurs at the centre of the simulated landscape, allowing the disease to spread in different directions. This design facilitates the transmission and avoids edge effects. The model runs for the equivalent of 1040 epidemiological weeks (20 epidemiological years). The first 520 weeks (10 epidemiological years) were used for population stabilisation, while the remaining weeks were focused on analysis. Furthermore, the state of each agent was updated in discrete time based on their interactions, allowing for a detailed simulation of infection spread over time. This approach provides insights into disease dynamics within each cell and across neighbouring cells. Finally, we collected metrics from each simulation, representing specific combinations of transmission rates and carrying capacity/autocorrelation. These metrics included agent populations, case counts, and deaths at each time point (epidemiological week).

### 2.3. Simulations

We propose three scenarios of NLMs/ABMs, building progressively and adding complexity to explore the effect of landscape and the model sensitivity to different parameters: (a) baseline homogeneous model (two species share a homogeneous landscape, meaning the carrying capacity is the same across all cells), (b) baseline heterogeneous model (two species share a heterogeneous landscape with a single level of autocorrelation), and (c) species-specific heterogeneous model (each species has its own overlapping heterogeneous landscape, in other words, two levels of autocorrelation are considered).

### 2.4. Baseline Models

#### 2.4.1. Homogenous Model

We considered ten distinct unitless carrying capacities (K), ranging from 0.1 to 0.9, with intervals of 0.09. As part of this scenario, we incorporated four different transmission rates. Previous studies about the intraspecies transmission between raccoon populations [39] have found a transmission rate of 0.035 as the standard value for rabies in Southern Ontario [61]. Thus, we selected values of 0.001 and 0.018 to represent minimal and lower transmission rates, and 0.052 as a higher value for each of three transmission rates: (1) the transmission rate from the reservoir species to the spillover species (spillover interspecies transmission rate; RS), (2) the transmission rate from the spillover species to the reservoir species (spillback interspecies transmission rate; SR), and (3) the transmission rate within individuals of the spillover species (spillover intraspecies transmission rate; SS). Notably, the reservoir intraspecies transmission rate was fixed at 0.035, as previously implemented [39].

To mitigate the influence of stochasticity inherent in the ABMs, we conducted 50 repetitions for each of the 640 landscapes (32,000 simulations). The number of replicates was predetermined and then assessed for suitability by comparing the accumulated median of cases to the final median. We deemed this number appropriate because the ratio of the accumulated median to the final median approached 1 (with a tolerance of 0.001) by the 11th replicate (IQR 1–32; range: 1–50; Appendix A).

#### 2.4.2. Heterogeneous Model

We considered ten distinct autocorrelation levels (H_0_), ranging from 0.1 to 0.9, with intervals of 0.09. Hence, we followed the same logic of the homogenous model for the transmission rates. To ensure that the landscape structure facilitated disease transmission, we selected landscapes where at least 60% of the cells had a minimum carrying capacity of 0.5. We performed 128,000 simulations obtained from 200 repetitions of 640 landscapes.

To assess the effect of landscape autocorrelation (H), which reflects the clustering or lack thereof of carrying capacities (K), we initially set the number of replicates for each simulated landscape to 100. This was based on the expectation that the relative median of new cases would stabilise with additional simulations, particularly when compared to the homogeneous landscape scenario. However, upon evaluating the adequacy of this repetition count, we observed a lack of stabilisation in some landscapes and decided to double it. After further analyses, we determined that 200 replicates were sufficient, meeting our predefined criteria for adequacy (relative median approximated to 1 around the 63rd replicate, IQR 26–126, range: 1–200; Appendix A).

### 2.5. Species-Specific Heterogeneous Model

The previous literature on habitat connectivity has highlighted the importance of identifying potential corridors for rabies spread [62,63,64,65]. These studies demonstrate how various environmental factors, such as land cover, soil type, and climate conditions, interact with landscape structure to influence the population dynamics of rabies transmission. Nevertheless, the rabies virus can jump to different species, and its distribution often aligns with that of its host species [39,61]. This inspired us to explore a two-species model across different landscape structures.

Based on the biological characteristics of raccoon rabies in North America, we made the following assumptions: (a) individuals of the reservoir species can only become infected through contact with other reservoir individuals; thus, spillback transmission is marginal with values for SR close to zero [38,66]; (b) as a reasonable premise reflecting the reduced efficiency of the viral strain in crossing host species, individuals of the spillover species are infected by reservoirs at half the transmission rate observed between reservoir individuals; (c) the spillover species has an intraspecies transmission rate ranging from 0.001 to 0.052; and (d) each species exhibits a unique level of landscape autocorrelation (H_1_ and H_2_), ranging from 0.1 to 0.9 in intervals of 0.009.

Following the approach used with the heterogeneous model, we chose landscapes where at least 60% of the cells had a carrying capacity of 0.5 or higher. This resulted in 80,000 simulations based on 200 repetitions across 400 landscapes. The framework used to determine the adequate number of replicates is the same as that applied to the baseline heterogeneous scenario (relative median approximated to 1 around the 96th replicate, IQR 46–162, range: 1–200; Appendix A).

Parameter combinations by type of model and landscape are shown in Table 1. More detailed tables with all the combinations explored are found in the Appendix A.

### 2.6. Output Analysis

The primary metric of interest was the median number of new reservoir and spillover cases across the time series for each landscape, defined by a combination of three transmission rates and levels of carrying capacity/autocorrelation. We selected the median instead of the mean cases because it is less sensitive to outliers, which could distort the interpretation of disease outbreaks, especially in scenarios with highly skewed distributions of cases [67,68]. We also chose to report the median across the entire time series rather than a time-specific measure (e.g., weekly or yearly), as this accounts for long-term trends and avoids potential bias from shorter-term fluctuations. Since multiple replicates were generated for each landscape, we first calculated the median value for each replicate and then computed the overall median for each landscape, allowing us to capture the central tendency of case numbers while minimising the influence of variability in individual replicates.

We plotted the results through heatmaps representing the median number of new cases. The horizontal axis indicates the landscape carrying capacity or autocorrelation (carrying capacities across cells in the grid), while the vertical axis represents transmission rates. For the baseline models, each figure contains three rows of heatmaps, one for each varying transmission rate (RS, SR, and SS). For the species-specific heterogeneous model, each row comprises a different spillover intraspecies transmission rate (SS) level. The colour scale ranges from the lowest to the highest median number of new infection cases. 

As a secondary metric, we examined the timestep (epidemiological week) at which the maximum number of new cases occurred for each replicate and then calculated the overall median across replicates for each landscape. To illustrate these results, we present Appendix A for the baseline and species-specific heterogeneous landscape models grouped by each combination of parameters.

## 3. Results

### 3.1. Baseline Models

#### 3.1.1. Homogeneous Model

Across the 640 homogeneous landscapes, the median number of reservoir infections was 924 (IQR 0–1664), while spillover infections had a median of 676 (IQR 0–1507). The landscape with the maximum median reservoir infections (2396) was distinguished by the highest carrying capacity (0.900), RS and SR rates (0.052), and the minimal SS rate (0.001). Similarly, the landscape with the greatest median spillover infections (2316) featured the highest carrying capacity (0.900), standard SR rate (0.035), and lower RS and SS rates (0.018).

Overall, simulations show that the median number of infections is higher in landscapes with cells with values of at least 0.3. Starting at this level of K, we obtained higher medians of cases by increasing transmission rates. Specifically, landscapes with a carrying capacity of 0.9 generally produced the highest median infections across all transmission rates. The impact of transmission rates on the median number of cases becomes more pronounced as the rate increases from 0.001 to 0.018, while the changes are less noticeable when the rate rises from 0.018 to 0.052 (Figure 1).

Regarding the median of new cases in the reservoir species, the RS rate seems to have a lower effect, as case numbers stay high even at very low values (see the K = 0.9 column in the heatmap). For the spillover species, the RS rate indicates that the number of cases is reduced at very low values. About the SR rate, there are fewer median cases in standard to higher rate settings, especially at increased levels of K. By contrast, a higher median of cases is determined at minimal to lower rates (0.001 and 0.018), with K ≥ 0.8. This applies to both the reservoir and the spillover species.

Lastly, the effect of the SS rate on the reservoir species depended on the landscape carrying capacity, with slight changes observed for moderate to high carrying capacities (see the K ≥ 0.633 columns in the heatmap). Nonetheless, this rate distinctly impacted the median infections in the spillover species, and there was a positive relationship between K and SS in the minimal to standard rates range.

The median time to reach the reservoir and spillover cases peak was 774 weeks (IQR 534–887) for both. Landscapes that achieved their peak infections most quickly were characterised by a combination of reduced carrying capacity and minimal to lower RS, SR, and SS rates. However, these landscapes generally produced very low overall case numbers, with many experiencing infection die-offs (Appendix A).

#### 3.1.2. Heterogeneous Model

In the 640 heterogeneous landscapes, the median reservoir infections were 1260 (IQR 977–1317), while median spillover infections reached 1231 (IQR 871–1304). The landscape with the highest median reservoir infections (1435) was notable for its maximum autocorrelation (0.900), standard SR and SS rates (0.035), and lower RS rate (0.018). Likewise, the landscape with the most spillover infections (1423) exhibited the maximum carrying capacity (0.900) and RS rate (0.052) yet had lower SR and SS rates (0.018).

Our results demonstrate that landscape autocorrelation (H_0_) influences the number of new infections, particularly as autocorrelation levels exceed 0.6. The median number of new infections increases noticeably in landscapes with higher autocorrelation. However, while landscape structure affects the median number of cases, transmission rates are also critical in determining the number of infections across different levels of H_0_ (Figure 2).

We found that minimal RS rates negatively affected new infection cases, particularly in landscapes with lower autocorrelation. This suggests that infection cases are unsuccessful in less connected patches with reduced transmission rates. Conversely, higher RS rates consistently led to more infection cases at all levels of H_0_. For the spillover species, minimal-to-lower RS rates (0.001 and 0.018) resulted in either no infections or fewer infections, indicating that reduced transmission rates were insufficient for the virus to spread effectively in these species across those levels of autocorrelation.

Concerning the SR rate, we show that minimal rates (0.001) are linked to a decrease in the median number of reservoir and spillover species cases. However, the reduction is more pronounced in the former. Moreover, the median number of new infections increased when transitioning from lower (0.018) to standard (0.035) rates, but there is the opposite effect when augmenting RS to its highest magnitude (0.052). Although landscape autocorrelation has an impact, it appears more subtle, with transitions occurring gradually along the purple-to-yellow scale.

The heatmap for the reservoir species appeared largely congruent regarding the SS rate, suggesting that SS has little to no effect on infection dynamics in this species. In contrast, there is a direct relationship between SS and median new infections in the spillover species. Likewise, the highest values of both H_0_ and the SS rate caused the greatest number of median cases for both the reservoir and spillover species.

The median duration to achieve the maximum reservoir and spillover infections was 781 weeks (IQR 727–889). Landscapes reaching their infection peaks most swiftly were defined by the highest levels of autocorrelation, RS, SR, and SS rates (Appendix A).

### 3.2. Species-Specific Heterogeneous Model

Among the 400 species-specific heterogeneous landscapes, the median reservoir infections were 697 (IQR 566–806), and spillover infections had a median of 858 (IQR 518–1263). The landscape with the greatest median reservoir infections (945) was marked by maximum autocorrelation in the reservoir landscape (0.900). In contrast, the landscape with the highest median spillover infections (1472) was characterised by the maximum autocorrelation in the spillover landscape and elevated SS rates. In this context, similar infection levels (1447–1462) were observed across the full spectrum of reservoir autocorrelation values.

Our approach reveals that landscape autocorrelation levels and infection rates modify the infection dynamics for both reservoir and spillover species. Spillover infection counts were more pronounced at higher transmission rates for the spillover species. This is illustrated in Figure 3, which presents two columns of heatmaps with the median number of infection cases. Each heatmap in this figure includes the landscape autocorrelation for the reservoir species (H_1_) on the horizontal axis and the landscape autocorrelation for the spillover species (H_2_) on the vertical axis.

In general, reservoir individuals within landscapes with higher levels of H_1_ are more likely to interact with each other due to the clustered distribution of carrying capacities (for all values of SS). This leads to an increase in the number of new cases within the reservoir species. In this scenario, the impact of H_2_ is minimal (i.e., no change on the vertical axis), and the reservoir species’ intrinsic dynamics primarily influence infection rates. The uniformity of colour within each column of H_1_ values is unaffected by variations in H_2_ because of the design of the model with marginal spillback.

The second column focuses on the median number of infections in the spillover species. Here, as in the first column, higher autocorrelation values of H_1_ contribute to increased infections within the spillover species. However, when the SS rate is ≥0.035, the number of cases in the spillover species becomes more strongly influenced by the levels of H_2_. The consistency of colour within each row of H_2_ values suggests that variation in H_1_ has little impact on the number of cases in standard to higher SS rates.

The median time to reach the peak reservoir infections was 943 weeks (IQR 937–992), while the spillover peak occurred at a median of 939 weeks (IQR 839–947). Landscapes achieving the fastest reservoir peaks were associated with maximum autocorrelation in the reservoir landscape. Meanwhile, those reaching the spillover infection peak most rapidly displayed a combination of maximum spillover autocorrelation, elevated SS rates, and a wide range of reservoir autocorrelation levels (0.367–0.900) (Appendix A).

## 4. Discussion

Our model, which combined NLMs and ABMs, showed that landscape autocorrelation influences infection patterns (particularly in the spillover species) but that transmission rates remain the primary factor driving infection dynamics. This underscores the importance of considering ecological context—through landscape features—and species-specific transmission dynamics when modelling the spread of infectious diseases in wildlife populations. In contrast to previous studies that have emphasised the role of landscape heterogeneity as a barrier to pathogen dispersal [15,69], particularly for highly virulent pathogens, our findings underline that transmission rates remain an influential determinant in infection dynamics. While landscape features can constrain host movement and limit pathogen spread to distant populations, a pathogen’s capacity to spread is primarily driven by the intrinsic transmission rates.

### 4.1. Animal Density and Spatial Distribution Across Landscapes

Homogeneous landscape simulations showed that the population density of susceptible species plays a critical role in facilitating virus transmission. In high-density populations, the proximity of individuals increases the likelihood of contact between infected and susceptible agents, thereby accelerating pathogen spread. Alternatively, while transmission may still occur in low-density areas, it is generally less effective and relies more on animal movement to facilitate exposure.

Research on animal density and disease transmission reveals unique relationships depending on pathogen dynamics. For instance, in regions with high livestock density, higher rates of acute respiratory infections and associated symptoms have been reported [70]. Similarly, in Tasmanian devils (*Sarcophilus harrisii*), the transmission of devil facial tumour disease has been linked to increased contact rates as population density rises, with disease spread being particularly frequency-dependent during the mating season [71]. This is consistent with the hypothesis that crowded conditions promote direct pathogen transmission. On the other hand, some studies reveal counterintuitive results for the prevalence of *Mycobacterium bovis* in European badgers, which was found to be higher in low-density populations and smaller social groups, suggesting that factors beyond density, such as social structure and contact frequency, may play more prominent roles in transmission [72]. Additionally, leptospira prevalence has been negatively associated with population density in house mice (*Mus musculus*), resulting in a relatively constant density of infected animals over time [73].

Although its dynamics may be density-dependent in the case of the rabies virus, recent studies have demonstrated that it can persist in heterogeneous landscapes, even when the mean carrying capacities are below the theoretical threshold observed in homogeneous environments [74].

### 4.2. Influence of Landscape Autocorrelation

In heterogeneous landscapes, we determined that the spatial distribution of carrying capacities impacted the spread of infection, especially in landscapes where autocorrelation values surpassed half the maximum. More autocorrelated areas exhibited a higher concentration of infections, as the proximity of agents within high-carrying-capacity zones facilitated transmission. These findings are consistent with previous research on *Borrelia burgdorferi*, which suggests that landscape connectivity influences the dispersion of Lyme disease-causing bacteria through its impact on host dispersal [75], and with studies showing that controlling wildlife corridors and habitat connectivity can help contain disease outbreaks, such as African swine fever [46]. Hence, we emphasise the value of combining connectivity and fragmentation analysis, using species distribution and accounting for the surface area and abundance of occupied habitat patches when managing disease spread.

Regarding the species-specific heterogeneous landscape model, our findings highlight that landscape autocorrelation and species-specific transmission rates influence the dynamics of new infections. While higher autocorrelation in the reservoir landscape can drive infections in both species, the impact of autocorrelation in the spillover landscape becomes more significant when the intraspecies transmission rate within the spillover species is sufficiently high. This indicates that landscape structure and transmission rates must be considered together to understand infection dynamics fully.

It is essential to note that spillover species’ vulnerability to zoonotic viruses is influenced by their population abundance, exploitation, and habitat loss [76], which can be summarised as landscape immunity [77]. To reduce these risks, efforts should prioritise enhancing landscape resilience with targeted measures, particularly in areas affected by human activities. Protected area management is key to preventing pandemics by ensuring the health of ecosystems, which ultimately supports the well-being of both wildlife and humans [21].

### 4.3. Cross-Species Transmission

We found that the highest spillback interspecies transmission rates (SR) and carrying capacities in our baseline model resulted in fewer infection cases. This is likely because the dynamics failed to adequately support the recovery of the susceptible population, which may be linked to high die-off rates. Multiple factors influence the transmission rate for spillover and spillback [78], including its efficiency and the cross-species barrier. Although viral mutations are frequent, only a few prevail in nature, indicating that successful cross-species transmission involves a complex evolutionary process.

Our results also consider that landscape autocorrelation and intraspecies transmission rate for the spillover species could increase the median number of cases. Rabies virus demonstrates a unique ability among lyssaviruses to establish infectious cycles in multiple host species [79]. This adaptability is attributed to complex evolutionary dynamics, including host shifts and global spread [80]. Experimental studies reveal that dog-adapted RABV can more easily replicate in foxes than vice versa, suggesting a unidirectional host evolutionary effect [81]. However, pathogen dynamics involve more than contact between susceptible-infected individuals [82]; for example, results of spatial and phylogenetic cluster analyses suggest that spillover and spillback transmission between raccoons and skunks is affected by topographical barriers [38,66].

Moreover, this model is based on dynamics observed in terrestrial mammals. When applying it to the order Chiroptera, which has been extensively studied in relation to rabies, the specific dynamics of bats must also be considered. In the context of rabies transmission using ABMs, it is crucial to consider the unique behavioural patterns and ecological factors that influence bat movements [83]. Bats are highly mobile, with some species capable of long-distance migration or daily foraging flights [84]. The model should incorporate key factors such as flight range, roosting behaviour, species diversity, uniqueness, and the endemicity of rabies within different ecosystems, as these variables may impact disease transmission dynamics in bat populations [85]. Indeed, a possible hypothesis would be that volant mammals are less sensitive to habitat structure, as they can traverse the territory more easily. This would manifest as a lowered or absent sensitivity to the landscape auto-correlation parameter.

Our study focuses on how viruses can spread across species, which has direct implications for how zoonotic diseases (like rabies) might spread from animals to humans. This model can be applied to address how spatial structure (e.g., urban vs. rural areas, proximity to wildlife) influences human exposure to pathogens. Our findings suggest that higher spatial autocorrelation in the presence of sufficient carrying capacity facilitates viral transmission, which could be applied to areas where high densities of humans and animals overlap. This could lead to more targeted prevention measures, such as vaccination of animals or surveillance of high-risk areas to detect early signs of zoonotic diseases.

### 4.4. Implications and Limitations for Modelling Infectious Diseases

The model’s primary strength lies in its integration of agent-based models (ABMs) and neutral landscape models (NLMs), providing a detailed simulation of pathogen spread that accounts for both spatial structure and agent interactions. This approach offers valuable insights into how landscape features, such as connectivity and agent density, influence viral transmission dynamics. In addition, considering both homogeneous and heterogeneous landscapes allows for exploring various ecological scenarios, enhancing its potential applicability.

However, the model has limitations, such as simplifying ecological interactions and using generalised parameters, which may not capture the full complexity of real-world systems. It also lacks empirical validation with field data, which could reduce its direct applicability to actual disease management scenarios. Future improvements involve calibrating and validating with real-world data and expanding the model to account for a broader range of species and landscape complexities.

We expect that our approach will become valuable for researchers and decision-makers to develop planned and data-informed strategies for managing outbreaks and mitigating spillover risk and that it can support more cost-effective and efficient interventions such as the following: (a) targeted vaccination in areas with high connectivity where the virus might spread more quickly, prioritising regions with dense wildlife populations, (b) monitoring and early warning systems, where the model could identify areas at higher risk for outbreaks, allowing for faster response times, and (c) habitat restoration or fragmentation efforts, which can be driven to immunologically competent communities that could handle disease transmission.

## 5. Conclusions

In conclusion, this study underscores the critical interplay between landscape structure, species-specific transmission rates, and infection dynamics in wildlife populations. While landscape autocorrelation and carrying capacities influence infection patterns, transmission rates remain the primary driver of disease spread. Our findings highlight the importance of a comprehensive approach integrating fragmentation analysis with ecological context and species-specific transmission dynamics. Effective disease management, particularly in the early stages of an outbreak, requires a deep understanding of these factors to design targeted interventions. Future research should focus on refining models with real-world landscapes and more precise transmission rate estimations to improve our ability to predict and manage the spread of infectious diseases in wildlife populations.

## Figures and Tables

**Figure 1 microorganisms-13-00416-f001:**
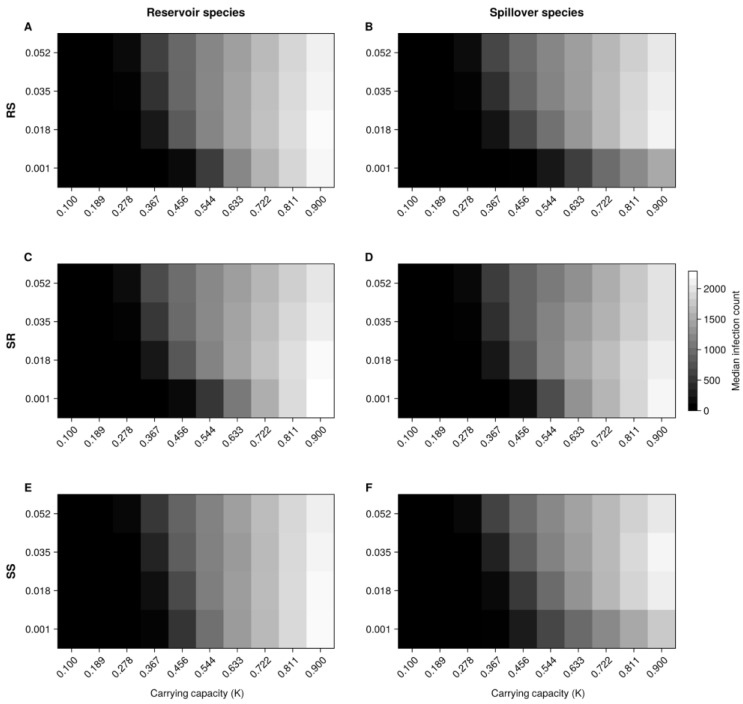
Effect of landscape carrying capacity (K) on transmission dynamics across species in homogeneous landscapes. The spillover (RS) (**A**,**B**) and spillback (SR) (**C**,**D**) interspecies transmission rates, as well as the intraspecies transmission rate in the spillover species (SS) (**E**,**F**), are displayed on separate vertical axes. The colour scale is based on the median number of new infections per epidemiological week in either the reservoir or spillover species.

**Figure 2 microorganisms-13-00416-f002:**
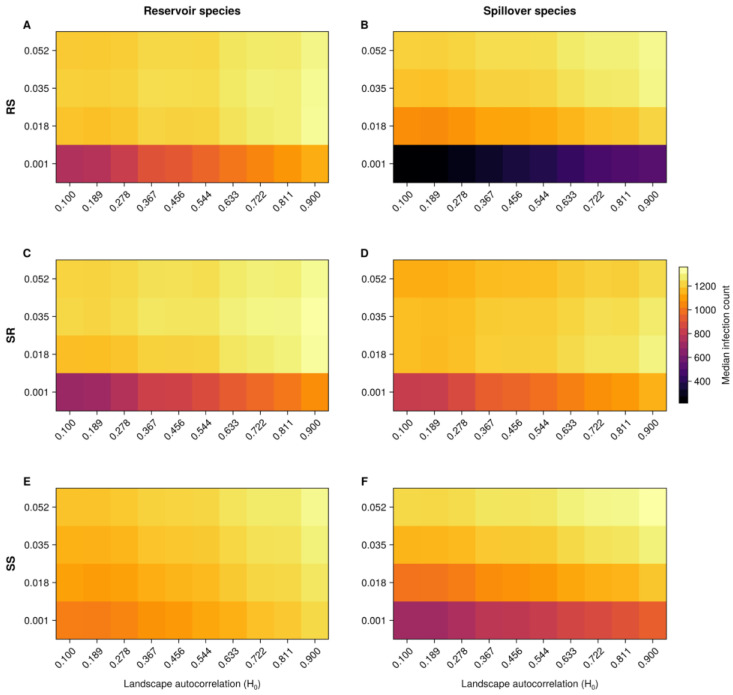
Effect of landscape autocorrelation (H_0_) on transmission dynamics across species. The spillover (RS) (**A**,**B**) and spillback interspecies (SR) (**C**,**D**) transmission rates, as well as the spillover intraspecies (SS) (**E**,**F**) transmission rate, are displayed on separate vertical axes. The colour scale is based on the median number of new infections in the reservoir or spillover species.

**Figure 3 microorganisms-13-00416-f003:**
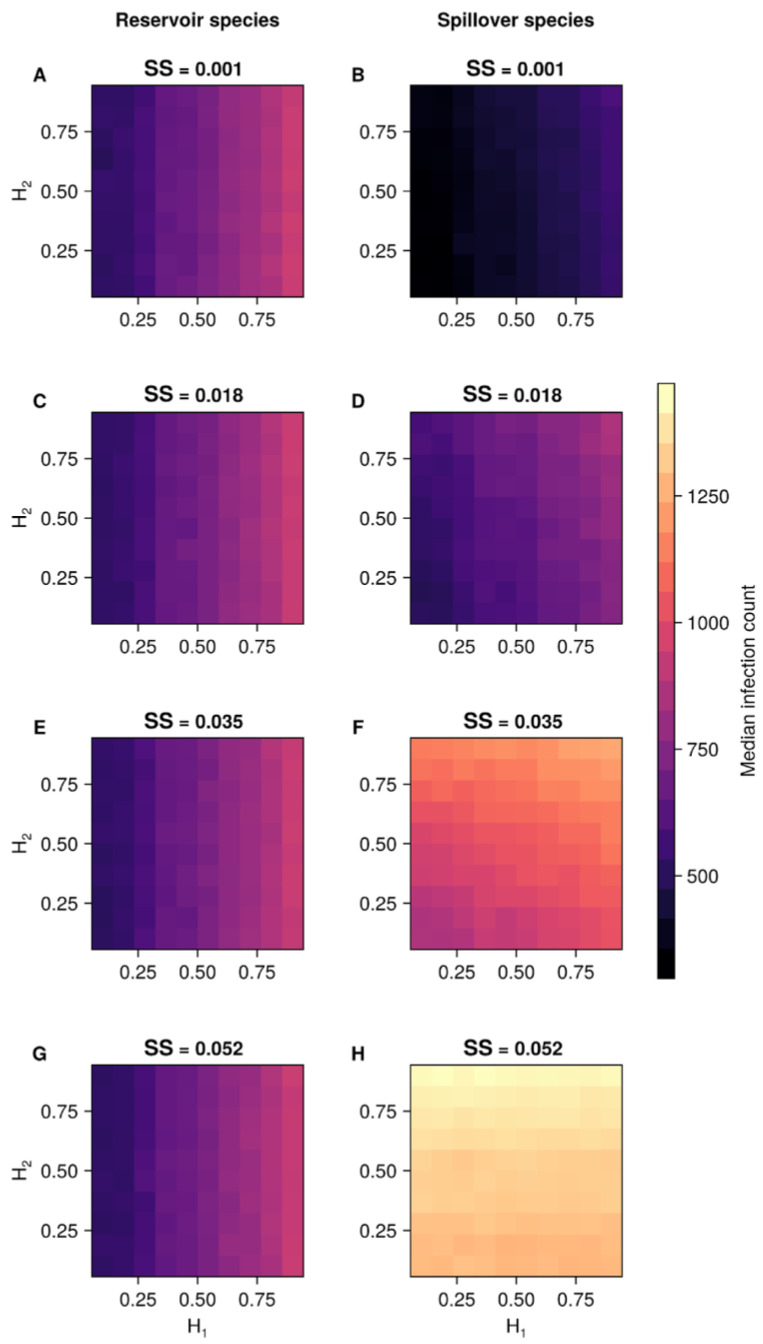
Effect of independent landscape autocorrelations (H_1_ for reservoir and H_2_ for spillover animals) on transmission dynamics across species. The colour scale is based on the median number of new infections in either the reservoir or spillover species, according to the intraspecies transmission rate within the spillover species (SS) (minimal (**A**,**B**), lower (**C**,**D**), standard (**E**,**F**), higher (**G**,**H**)).

**Table 1 microorganisms-13-00416-t001:** Simulation parameters according to the type of model. These NLMs use cells that are not tied to real-world units. This design provides a framework to measure how landscape spatial structure influences outbreak dynamics. Transmission rates represent the probability of rabies transmission during the contact between an infected and a non-infected individual. Parameters are based on previous studies [38,64]. The model runs for 1040 epidemiological weeks.

Model	Intraspecies Rates		Interspecies Rates		Spatial Structure	
	RR	SS	RS	SR	K	H
Baseline model: homogeneous landscape ^a^	0.035	0.001, 0.018, 0.035, 0.052	0.001, 0.018, 0.035, 0.052	0.001, 0.018, 0.035, 0.052	0.1–0.9 (steps 0.09)	-
Baseline model: heterogeneous landscape ^b^					0.1–0.9 (60% with ≥0.5)	H_0_: 0.1–0.9 (steps 0.09)
Species-specific heterogeneous model ^c^	0.035	0.001, 0.018, 0.035, 0.052	0.018	0.001	0.1–0.9 (60% with ≥0.5)	H_1_: 0.1–0.9 (steps 0.09) H_2_: 0.1–0.9 (steps 0.09)

RR (reservoir–reservoir); SS (spillover–spillover); RS (reservoir–spillover): spillover phenomenon; SR (spillover–reservoir): spillback phenomenon; K (carrying capacity), H (spatial autocorrelation): H_0_ (homogeneous landscape), H_1_ (heterogeneous landscape: reservoir), H_2_ (heterogeneous landscape: spillback). ^a^ NLMs are characterised by matrices with a homogeneous distribution across the landscape. All cells have identical values, representing a uniform spatial structure. Both species share the same landscape (K). ^b^ Each landscape incorporates varying degrees of H, with autocorrelation affecting the degree of similarity between the K values of neighbouring cells. Both species share the same autocorrelated landscape (H_0_). ^c^ Each landscape incorporates varying degrees of H, with autocorrelation affecting the degree of similarity between the K values of neighbouring cells. Each population has its autocorrelated landscape (H_1_ and H_2_).

## Data Availability

The code to generate the NLMs and ABMs, as well as results from simulations, are available at https://github.com/Norma-Forero/abm_rabies (accessed on 19 January 2025). Details about the NeutralLandscapes.jl package can be found at https://github.com/EcoJulia/NeutralLandscapes.jl (accessed on 15 September 2024).

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
