# Peer review of "Spatial Landscape Structure Influences Cross-Species Transmission in a Rabies-like Virus Model"

_microorganisms, 2025, doi:10.3390/microorganisms13020416_

Round 1

Reviewer 1 Report

Comments and Suggestions for Authors

The authors aimed to simulate biologically realistic agent-based models over neutral landscapes to examine how spatial structure affects the spread of a rabies-like virus in a two-species system. Building landscapes with varying autocorrelation levels and simulated disease dynamics using different transmission rates for intra- and interspecies spread. Analysis was based on combinations of spatial landscape structures and transmission rates, focusing on the median number of new reservoir and spillover cases.

It has been a pleasure to review a manuscript of this quality, not just because of the detailed information on material and methods but also because of the topic that is evaluated within the study. It is essential to understand how the landscape could affect the dynamics of infectious diseases and ecology.

I have one issue with this: the specific example of rabies. In this case, the study has limitations related to the considered reservoir species, and the scenario will be completely different if bats are considered the reservoir. The landscape effect would be different, which should have been discussed further in the manuscript.

Author Response

We thank the reviewer for the insightful suggestion, as it is very important to consider different rabies hosts when using ABM. We have discussed further this issue in the manuscript (see lines 477-488, highlighted in yellow):

Moreover, this model is based on dynamics observed in terrestrial mammals. When applying it to the order Chiroptera, which has been extensively studied in relation to rabies, the specific dynamics of bats must also be considered. In the context of rabies transmission using ABMs, it is crucial to consider the unique behavioural patterns and ecological factors that influence bat movements (84). Bats are highly mobile, with some species capable of long-distance migration or daily foraging flights (85). The model should incorporate key factors such as flight range, roosting behaviour, species diversity, uniqueness, and the endemicity of rabies within different ecosystems, as these variables may impact disease transmission dynamics in bat populations (86). Indeed, a possible hypothesis would be that volant mammals are less sensitive to habitat structure, as they can traverse the territory more easily. This would manifest as a lowered or absent sensitivity to the landscape auto-correlation parameter.

Reviewer 2 Report

Comments and Suggestions for Authors

I have finished reviewing the manuscript entitled "Spatial landscape structure influences cross-species viral transmission" it is a novel well written manuscript.

In certain sections it is difficult to interpret if what you mean with "individuals" relates to humans or hosts or veterinary species, it should be clearly specified along the manuscript. Such is the case in line 16, if you refer tu human cases, then it should be stated as incidence instead of "highest number of cases".

provide citations and references for the strategy you describe in lines 112-118

Line 140. Did you include years with 53 epidemiological weeks? it is difficult not to consider them in a 10-year period.

Please replace reference 34 for a more trustable source, it has a broken link.

Author Response

We thank the reviewer for the insightful suggestions. We discussed this issue further in the manuscript and made the necessary changes:

We referred to individuals as animals (agents in the models) (see lines 23, 29, and 134-135).

We provided references for the strategies described (see line 127).

We clarified the information regarding epidemiological weeks (see lines 153 and 154).

We rectified the reference link for 34.   Weldon C, Channing A, Misinzo G, Cunningham AA. Disease driven extinction in the wild of the Kihansi spray toad (Nectophrynoides asperginis) [Internet]. Epidemiology; 2019 [cited 2024 Dec 22]. Available from: http://biorxiv.org/lookup/doi/10.1101/677971

Reviewer 3 Report

Comments and Suggestions for Authors

Thank you for the review opportunity!

The manuscript received for review is well written.

Here are my (very minor) suggestions and comments:

Title: i would add the "rabies-like viruses" somewhere in the title; otherwise it does not make much sense what the study refers to.

Abstract: well written, nothing much to add.

Introduction: well written, nothing to add, extremely detailed with up-to-date references.

Materials and methods: ok, nothing to add.

Results: well presented.

Discussions: i would add the limitations as well in a different sub-character.

I would add a paragraph on how this research may be translated to humans.

Conclusions: nothing to add.

I recommend accepting this article with very minor changes.

Author Response

We thank the reviewer for the insightful suggestions. We have discussed this issue further in the manuscript and made the changes needed, which are highlighted in yellow:

We changed the title of the paper for Spatial landscape structure influences cross-species transmission in a rabies-like virus model.

We present the limitations in a new paragraph (see lines 498 and 506-511, highlighted in green):

However, the model has limitations, such as simplifying ecological interactions and using generalized parameters, which may not capture the full complexity of real-world systems. It also lacks empirical validation with field data, which could reduce its direct applicability to actual disease management scenarios. Future improvements involve calibrating and validating with real-world data and expanding the model to account for a broader range of species and landscape complexities.

We added a paragraph about the opportunities of this research for humans (see lines 489-498, highlighted in yellow):

Our study focuses on how viruses can spread across species, which has direct implications for how zoonotic diseases (like rabies) might spread from animals to humans. This model can be applied to address how spatial structure (e.g., urban vs. rural areas, proximity to wildlife) influences human exposure to pathogens. Our findings suggest that higher spatial autocorrelation in the presence of sufficient carrying capacity facilitates viral transmission, which could be applied to areas where high densities of humans and animals overlap. This could lead to more targeted prevention measures, such as vaccination of animals or surveillance of high-risk areas to detect early signs of zoonotic diseases.